# Ternary structure of the outer membrane transporter FoxA with resolved signalling domain provides insights into TonB-mediated siderophore uptake

Inokentijs Josts[1,2]*, Katharina Veith[1,2], Henning Tidow[1,2]*

[1]The Hamburg Centre for Ultrafast Imaging, University of Hamburg, Hamburg, Germany; [2]Department of Chemistry, Institute for Biochemistry and Molecular Biology, University of Hamburg, Hamburg, Germany

**Abstract** Many microbes and fungi acquire the essential ion $Fe^{3+}$ through the synthesis and secretion of high-affinity chelators termed siderophores. In Gram-negative bacteria, these ferric-siderophore complexes are actively taken up using highly specific TonB-dependent transporters (TBDTs) located in the outer bacterial membrane (OM). However, the detailed mechanism of how the inner-membrane protein TonB connects to the transporters in the OM as well as the interplay between siderophore- and TonB-binding to the transporter is still poorly understood. Here, we present three crystal structures of the TBDT FoxA from *Pseudomonas aeruginosa* (containing a signalling domain) in complex with the siderophore ferrioxamine B and TonB and combine them with a detailed analysis of binding constants. The structures show that both siderophore and TonB-binding is required to form a translocation-competent state of the FoxA transporter in a two-step TonB-binding mechanism. The complex structure also indicates how TonB-binding influences the orientation of the signalling domain.
DOI: https://doi.org/10.7554/eLife.48528.001

*For correspondence:
josts@chemie.uni-hamburg.de (IJ);
tidow@chemie.uni-hamburg.de (HT)

**Competing interests:** The authors declare that no competing interests exist.

## Introduction

Iron is one of the most abundant elements on earth and is essential for life. However, the bioavailability of iron in the environment is extremely low, and under aerobic conditions iron is found mostly as insoluble hydroxides. The demand for the ionic form of iron by all microorganisms growing in iron-limited conditions has led to the evolution of several efficient iron scavenging strategies. One of the predominant mechanisms by which microbes and fungi acquire iron is through the synthesis and secretion of small, specific high-affinity chelators termed siderophores, which keep iron in a chelated, soluble state (*Neilands, 1995*). In Gram-negative bacteria, these ferric-siderophore complexes are actively taken up into cells using highly specific TonB-dependent transporters (TBDTs) situated in the bacterial outer membrane (OM) as well as specific transporters present in the bacterial inner membrane (IM) (*Ferguson and Deisenhofer, 2004*). The energy for this uptake process is derived from the proton motive force and relies on the energising complex consisting of TonB/ExbB/ExbD situated in the bacterial IM (*Celia et al., 2016*; *Noinaj et al., 2010*). TonB acts as a physical link between the transporters in the OM and the energising complex in the IM (*Jordan et al., 2013*). Siderophore binding facilitates TBDT contact with the C-terminus of TonB through an allosteric mechanism, which exposes the TonB-binding site within TBDT, known as the TonB box, to the periplasm (*Lundrigan and Kadner, 1986*). Association of TBDTs with TonB establishes the main point of contact with ExbB/ExbD and the proton motive force provides the energy needed for the translocation of siderophores through the lumen of the OM barrel. The precise mechanism of this

translocation process is not yet fully understood but is thought to involve a mechanical extraction or unfolding of the plug region from within the barrel lumen (*Hickman et al., 2017*). Furthermore, a subclass of TBDTs possesses an N-terminal signalling domain, which regulates gene transcription of target operons, often participating in siderophore uptake and processing (*Enz et al., 2000*). The activation of these signalling cascades is both ligand- and TonB-dependent, however the molecular details of this signalling process and its activation remain highly elusive (*Ferguson et al., 2007*; *Koebnik, 2005*). It is speculated that the N-terminal pocket of the signalling domain is involved in the interactions with the σ-factor regulator proteins. To date, the crystal structures of the intact TBDT FpvA with the fully-resolved signalling domain suggest that there is a high degree of flexibility at the N-terminal region of the transporter (*Brillet et al., 2007*). However the putative site involved in contacting the regulator protein is tucked away beneath the barrel lumen. *Pseudomonas aeruginosa* is a Gram-negative bacterium and an opportunistic human pathogen, which is a major cause of hospital-acquired infections in immunocompromised patients. In patients with cystic fibrosis *P. aeruginosa* lung infections are usually associated with increased mortality rates. *P. aeruginosa* is able to utilise a range of xenosiderophores, that is siderophores produced by other bacteria or fungi in order to scavenge free iron. Such instances of so-called 'siderophore piracy' highlight bacterial adaptability and potential for colonising in an extremely broad range of environmental niches. One example of siderophore piracy is the uptake of ferrioxamine B, a hydroxamate siderophore produced by many *Streptomyces* species, by the specific OM transporter FoxA (*Llamas et al., 2006*). FoxA belongs to the family of TBDTs (transducers) and is involved in ferrioxamine B transport as well as modulation of transcriptional cascades in the bacterial cell. Ferrioxamine B uptake comes at a relatively low energetic cost, when compared with the production of native siderophores such as pyoverdin and pyochelin (*Dumas et al., 2013*). Indeed, when grown in the proximity of *Streptomyces ambofaciens*, several *Pseudomonas* species do not produce their own siderophores and instead parasitize on the siderophores of their neighbour by expressing the ferrioxamine B transporter, FoxA (*Galet et al., 2015*).

Here, we determined several crystal structures of FoxA from *P. aeruginosa*, in the apo state as well as in complex with the siderophore ferrioxamine B and TonB. Using a hybrid approach combining X-ray crystallography with biophysical interaction studies, we provide insights into TonB-mediated siderophore uptake across the bacterial OM as well as TonB-dependent signalling. Our results indicate that the transporter can exist in several different conformations, and that both substrate- and TonB-binding is required to form a translocation-competent state of the FoxA transporter in a two-step TonB-binding mechanism necessary for transport function.

## Results and discussion

### Ternary structure of FoxA bound to ferrioxamine B and $TonB_{Ct}$

Active uptake of siderophores such as ferrioxamine B across the outer membrane (OM) relies on the establishment of a physical contact between the specific transporter and inner membrane (IM)-tethered TonB. In order understand the mechanism of complex formation between FoxA and TonB we determined the crystal structure of the ternary complex consisting of FoxA bound to ferrioxamine B and the C-terminal TonB domain (residues 251-340, referred to hereafter as $TonB_{Ct}$). Two complexes were present in the asymmetric unit, with crystal contacts generated through the exposed soluble regions of both proteins (*Figure 1—figure supplement 1A,B*). We could resolve almost an entire FoxA molecule including the signalling domain (residues 53-820, with 11 amino acids missing from the N-terminus after the signal peptide) as well as the full $TonB_{Ct}$. We can identify two main modes of contact between FoxA and $TonB_{Ct}$. Similar to the FhuA and BtuB-TonB complexes (*Pawelek et al., 2006*; *Shultis et al., 2006*) the primary interaction site between FoxA and $TonB_{Ct}$ occurs through β-augmentation with parallel strands forming between residues 332-337 of TonB and residues 142-146 of FoxA (*Figure 1A,B*). In addition to the observed β-augmentation, the unstructured polypeptide segment upstream of the TonB box (residues 135-141) forms complementary contacts with the surface of TonB molecule mediated by backbone H-bond interactions as well as sidechains $Leu_{137}$, $Met_{139}$ and $Val_{142}$ located in a small cavity on the surface of TonB (*Figure 1C*). Most of the TBDTs do no harbour this extra binding motif upstream of the TonB box since it would only be present in TBDTs involved in regulating signalling events at the OM via the additional N-terminal

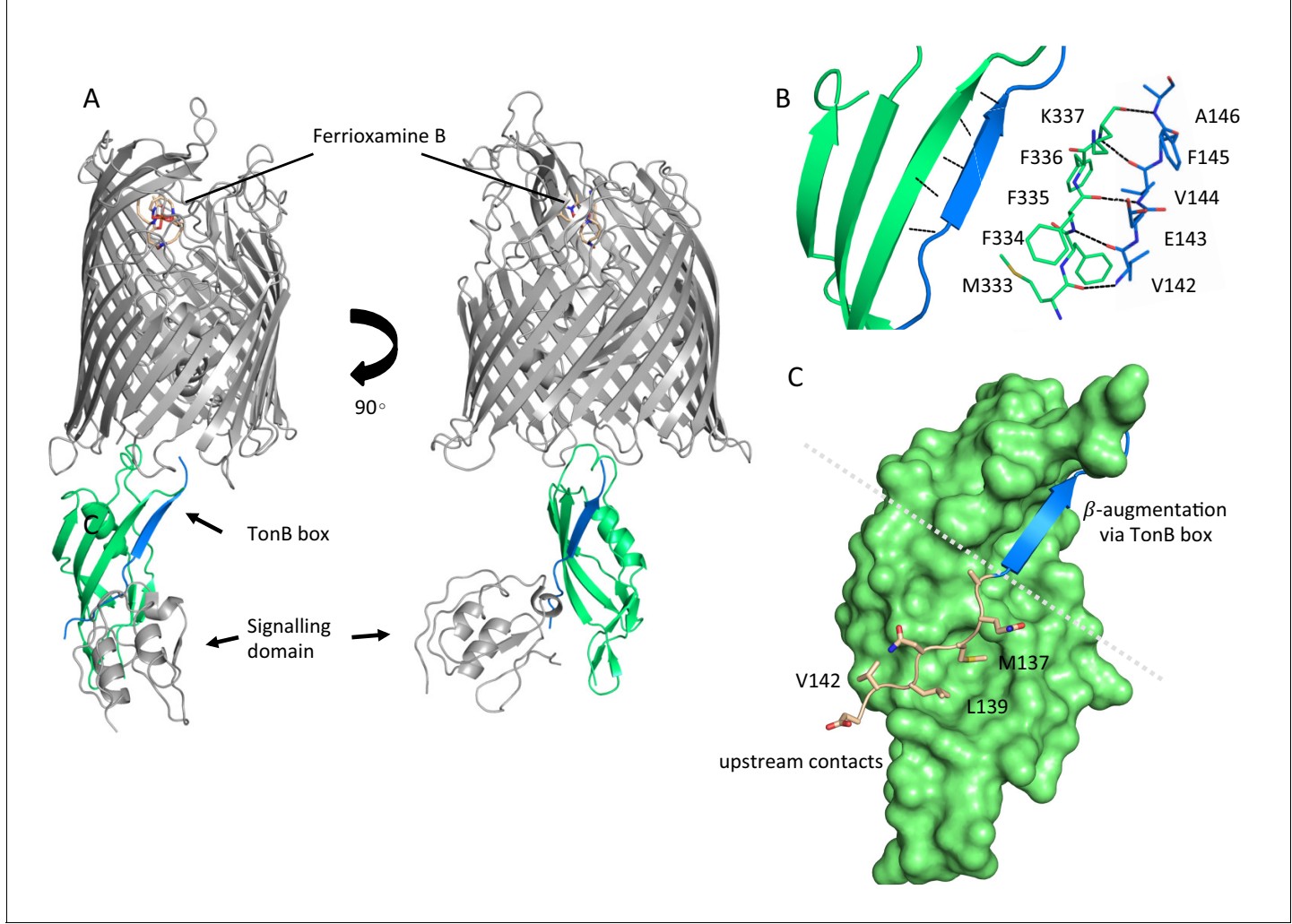

**Figure 1.** Complex formation between ferrioxamine B-bound FoxA and TonB is driven by multiple binding sites. (**A**) Overview of the ferrioxamine B-bound FoxA-TonB$_{Ct}$ complex. TonB$_{Ct}$ (green) interacts with the TonB box (blue) of FoxA (grey). (**B**) Parallel β–strands formed between the TonB box of FoxA (blue) and TonB$_{Ct}$ (green) through β-augmentation. All contacts are mediated predominantly by backbone hydrogen bonds between the two proteins. (**C**) Polypeptide stretch (pink) upstream of the TonB box (blue) forms additional contacts with the surface of TonB (green).

DOI: https://doi.org/10.7554/eLife.48528.002

The following figure supplements are available for figure 1:

**Figure supplement 1.** Crystal packing of the FoxA-ferrioxamine B-TonB$_{Ct}$ complex.

DOI: https://doi.org/10.7554/eLife.48528.003

**Figure supplement 2.** Flexibility in the FoxA-TonB complex.

DOI: https://doi.org/10.7554/eLife.48528.004

**Figure supplement 3.** Comparison of different TBDT-TonB complex structures reveals distinct mechanisms of TonB capture and positioning.

DOI: https://doi.org/10.7554/eLife.48528.005

domain in the periplasm. The secondary point of contact involves the side chains of residues in the periplasmic loops and barrel interior of FoxA with several residues in the loops of TonB facing the barrel. These contacts are mediated by electrostatic forces between TonB$_{R271}$ and FoxA$_{D352}$, and between TonB$_{R266}$ and FoxA$_{D355}$ as well as a side-chain (FoxA$_{E316}$) to the backbone carbonyl (TonB$_{S300}$) via a hydrogen bond (*Figure 1—figure supplement 2*). These contacts provide a secondary site of attachment for the TonB fragment and tether the C-terminal region of TonB to the barrel. This tethering locks the orientation of TonB against the barrel and the membrane plane. An overlay of the two complexes found in the asymmetric unit reveals that TonB$_{Ct}$ and the signalling domain are rotated by 9° with respect to the barrel and the membrane plane (*Figure 1—figure supplement*

2). It is evident that TonB$_{Ct}$ and the signalling domain experience some degree of flexibility in the distal part of the complex, whereas the proximal part of the complex is most likely stabilised by the contacts at the secondary site. In the previous structural models (*Pawelek et al., 2006*; *Shultis et al., 2006*), TonB sits in close proximity to the β-barrel, almost parallel to the putative lipid bilayer plane. In our crystal structure, the TonB fragment is located almost perpendicular to the β-barrel and the membrane plane highlighting the potentially dynamic nature of TBDR-TonB complexes, which has been suggested by recent EPR experiments (*Sarver et al., 2018*) (*Figure 1—figure supplement 3*). Overall, the structure of the ternary FoxA-ferrioxamineB-TonB$_{Ct}$ complex presented in this work reveals both the structure of the N-terminal signalling domain as well as a markedly different orientation of the TonB$_{Ct}$ relative to the TBDT compared to previously determined ternary structures of FhuA and BtuB (*Pawelek et al., 2006*; *Shultis et al., 2006*).

To understand the conformational changes occurring in FoxA in response to ferrioxamine B and TonB$_{Ct}$ we also determined the crystal structure of FoxA in its apo state (*Figure 2A*). In this structure a large solvent-exposed lumen faces the extracellular side of the membrane and is filled with solvent molecules. No electron density was present for the last two amino acids of the TonB box region and the N-terminal signalling domain of FoxA (residues 45-143), most likely due to the high flexibility of the linker connecting the plug domain and the signalling domain, as previously observed in the structures of FecA (*Ferguson et al., 2002*; *Yue et al., 2003*) and FpvA (*Brillet et al., 2007*; *Greenwald et al., 2009*; *Wirth et al., 2007*). Inspection of both apo as well as ligand- and TonB$_{Ct}$-bound crystal structures of FoxA revealed that in the apo state the TonB box is predominantly occluded in the interior of the barrel domain. A comparison of the plug domain conformations in both of our FoxA structures indicates that the TonB box must be displaced by approximately 22 Å from the folded plug domain in order to allow for β-augmentation to occur with TonB (*Figure 2B*, *Video 1*).

Compared to the apo FoxA structure the ternary complex of FoxA-FoaB-TonB$_{Ct}$ also reveals substantial loop movements. Displacement of loops 7 and 8 by approx. 7 Å on the extracellular side of the membrane leads to the closure of the barrel lumen on both sides of the membrane, limiting the access to the barrel lumen (*Figure 2C*, *Video 1*). Mechanistically, this would prevent the dissociation of the siderophore during translocation and opening of the entry channel within the barrel.

## Biophysical characterisation of the interactions between FoxA and TonB$_{Ct}$

Previous structural and biochemical investigations into the mechanisms of TBDT activation and substrate uptake have shown that siderophore binding usually leads to an unwinding of either an N-terminal helix or a stretch of polypeptide within the plug domain bearing the TonB box motif. This mechanism of polypeptide unwinding, initiated by concerted small motions throughout the plug, allows the C-terminus of TonB to make contact with the loaded transporter molecule. Moreover, insights into TBDT association with TonB paint a very complex, and often conflicting picture of substrate-dependent transporter activation and TonB binding. One model suggests a constitutively bound TonB-TBDT complex (*Adams et al., 2006*; *Kim et al., 2007*), whilst another proposes a cooperative mode of transporter-TonB interactions that is driven by initial substrate capture by the TBDT (*Cadieux and Kadner, 1999*; *Merianos et al., 2000*).

Therefore, we sought to understand the nature of FoxA-TonB interactions using isothermal titration calorimetry (ITC) to characterise the thermodynamics of FoxA-TonB interactions. For this purpose, we have reconstituted FoxA into MSP1D1 nanodiscs in order to minimise the detergent mismatch (*Fanucci et al., 2003*; *Mills et al., 2014*). Additionally, nanodiscs (ND) provide a lipidic scaffold and alleviate the deleterious effects detergents might have on the conformation of the transporter. Titration of TonB$_{Ct}$ into apo FoxA-ND complexes resulted in strong, saturable exothermic heats indicative of protein association. ITC data were fitted to a single binding site model and yielded a K$_d$ of 111 ± 6 nM. The binding process is enthalpically driven with a large, negative ΔH of -10.1 ± 0.8 kcal mol$^{-1}$ (*Figure 3A* and *Supplementary file 1*). No binding was observed when TonB$_{Ct}$ was titrated into empty nanodiscs. The observation of tight binding between TonB$_{Ct}$ and apo FoxA in nanodiscs is in contrast to similar experiments performed with FhuA and TonB$_{Ct}$, for which no binding could be observed (*Mills et al., 2014*). We speculate that the presence of the N-terminal domain and the interacting region upstream of the TonB box in FoxA is responsible for the differences in the binding modes between these two transporters.

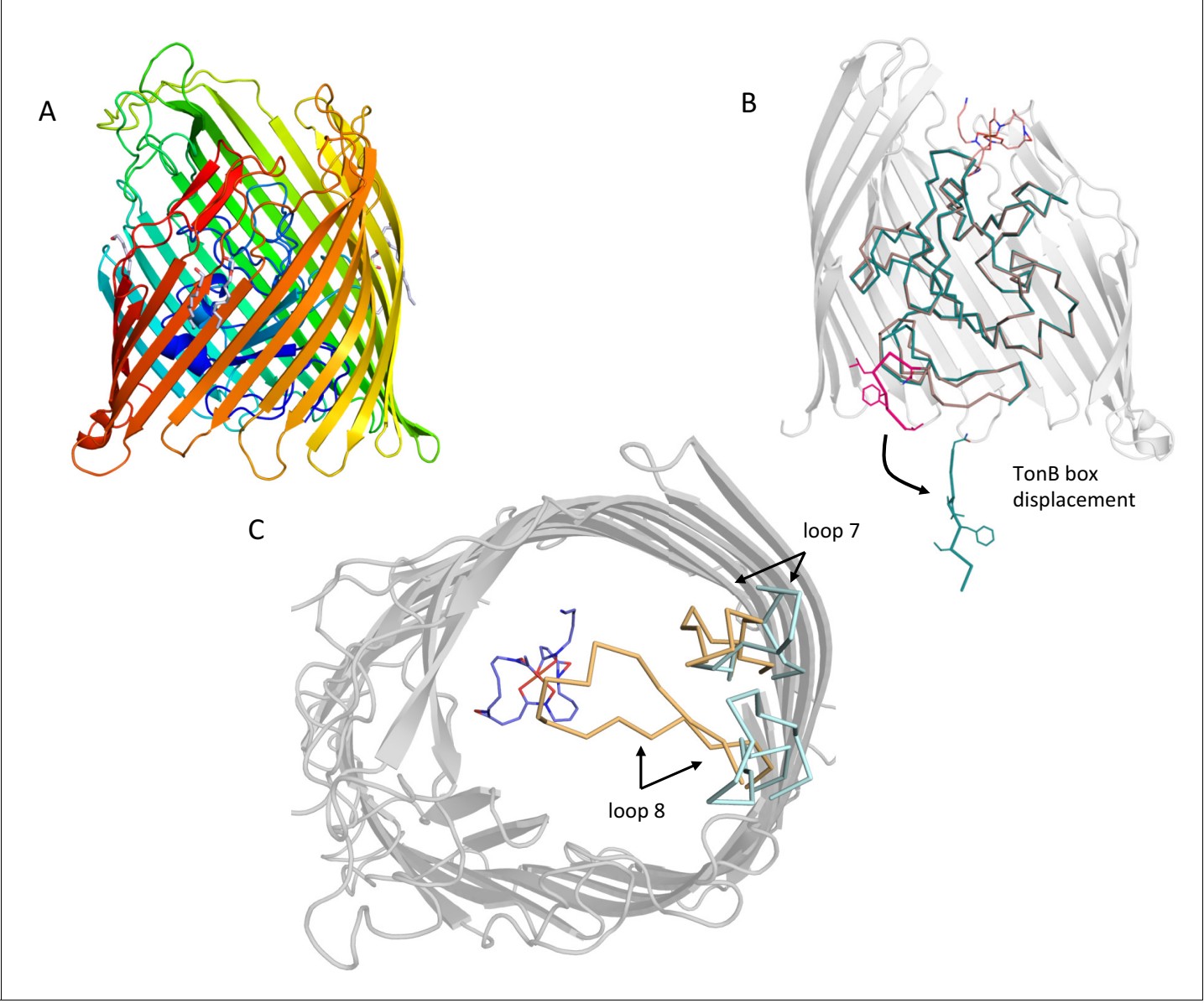

**Figure 2.** Conformational changes in the plug domain and extracellular loops of FoxA in response to ferrioxamine B and TonB binding. (**A**) The overall fold of apo FoxA consists of a 22-stranded β-barrel lined by the small globular plug domain within the lumen. The structure is colour-coded from blue (N-terminus) to red (C-terminus). (**B**) Structural rearrangements within the plug domain necessary to accommodate interactions with $TonB_{Ct}$. The region of the polypeptide being part of the TonB box common to both FoxA structures is highlighted in pink. This region is displaced by approximately 22 Å into the periplasm. Slight conformational changes are also observed throughout the rest of the plug domain (blue: ternary complex/brown: apo state). (**C**) Loops 7 and 8 enclose the bound siderophore within the hydrophobic cavity to prevent its dissociation and reduce permeation across the bacterial membrane during the process of siderophore uptake. Loop closure is only evident once the FoxA is bound with ferrioxamine B and the $TonB_{Ct}$ fragment (loops coloured brown), indicative of allosteric communication between the extracellular and periplasmic regions of the transporter (blue loops correspond to the apo FoxA).

DOI: https://doi.org/10.7554/eLife.48528.006

Next, we analyzed the interactions between $TonB_{Ct}$ and ferrioxamine B-bound FoxA in lipid nano-discs. The data were also fitted to a single binding site model. Our ITC experiments showed that in the presence of ferrioxamine B the binding affinity is increased 17-fold yielding a $K_d$ value of 6.6 ± 1.2 nM. The thermodynamic parameters of the association reaction also differ such that the ΔH value decreases drastically to -18.1 ± 0.9 kcal mol$^{-1}$ with an entropically unfavourable contribution of TΔS = -6.62 kcal mol$^{-1}$ (*Figure 3B* and *Supplementary file 1*).

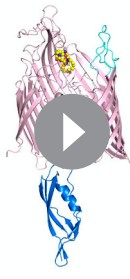

**Video 1.** Visualising the conformational changes occurring in FoxA in response to ferrioxamine B and TonB$_{Ct}$ binding. We observe the closure of extracellular loops 7 and 8 as a prerequisite for translocation of ferrioxamine B. At the periplasmic side, TonB box is expelled from the plug domain in order to make contacts with the TonB molecule.
DOI: https://doi.org/10.7554/eLife.48528.007

We speculate that the decrease in the entropy of TonB$_{Ct}$ binding in the presence of ferrioxamine B most likely arises from large conformational restrictions of the flexible and highly mobile TonB and signalling domains as well as folding/desolvation events involved in association with TonB$_{Ct}$. The difference in thermodynamics between the two binding reactions also suggests several distinct binding modes between TonB$_{Ct}$ and FoxA, which rely on the siderophore capture. The negative entropy is compensated for by a large decrease in the enthalpy of binding, which is driving the association reaction. Our ITC data suggest that in the presence of ferrioxamine B, FoxA is able to form a much tighter complex with TonB$_{Ct}$. Since β-augmentation between the TonB box of FoxA and TonB$_{Ct}$ is driven predominantly by hydrogen-bonded interactions, the stark decrease in the enthalpy of binding in our ITC could be explained by the formation of these additional contacts. The increased affinity and drastically reduced enthalpy of association is indicative of a much larger surface area participating in the complex formation process compared with the thermodynamics of apo FoxA-TonB$_{Ct}$ complexes. Altogether, the interaction studies presented here strongly support a cooperative mechanism of siderophore-dependent TonB capture by FoxA and that two distinct TonB-binding events can occur at the FoxA transporter.

To delineate the interactions between FoxA and TonB we generated FoxA variants with truncations in the signalling domain and analysed complex formation using analytical size-exclusion chromatography. Full-length FoxA exhibits three distinct elution profiles corresponding to apo protein, FoxA-TonB$_{Ct}$ complex and the ternary complex FoxA-ferrioxamine B-TonB$_{Ct}$ confirming our ITC

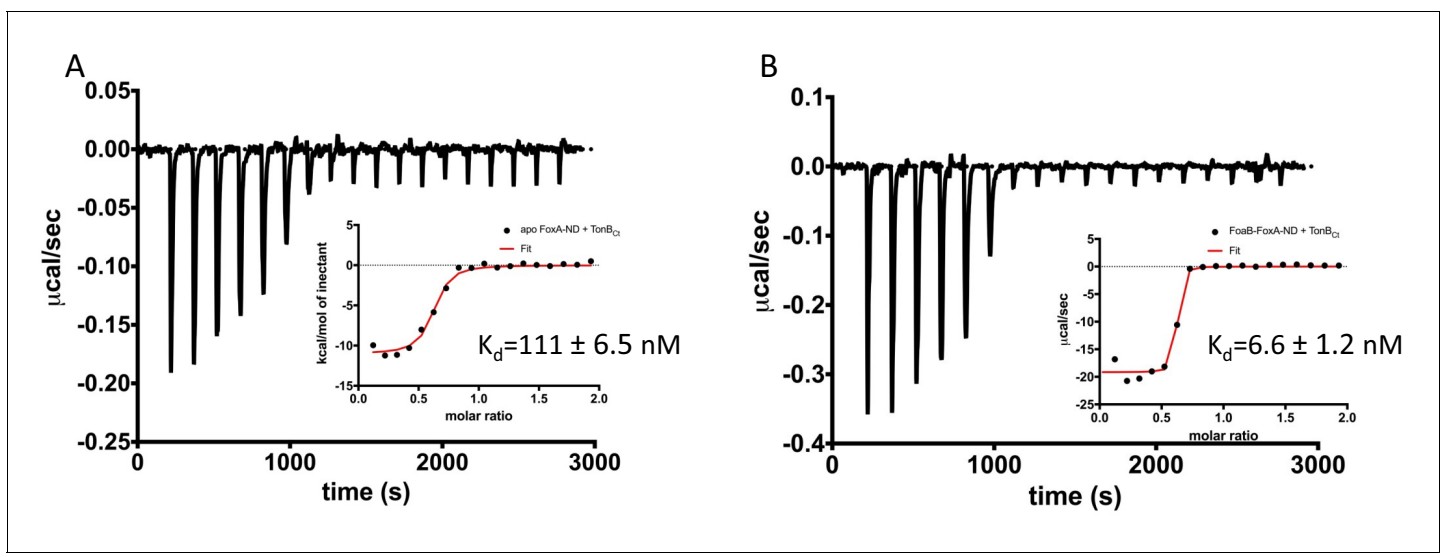

**Figure 3.** Two distinct modes of association between FoxA and TonB as revealed by the thermodynamics of complex formation. ITC profiles showing titration of TonB$_{Ct}$ (150 µM) into 15 µM apo FoxA nanodisc complexes (**A**) and 15 µM ferrioxamine B-FoxA nanodisc complexes (**B**). Insets show the integrated heats of binding with a single-site fit to the data.
DOI: https://doi.org/10.7554/eLife.48528.008
The following figure supplement is available for figure 3:

**Figure supplement 1.** Delineation of the constitutive TonB-binding motif in FoxA using analytical size-exclusion chromatography.
DOI: https://doi.org/10.7554/eLife.48528.009

findings of two distinct TonB$_{Ct}$-bound states, with and without the siderophore (*Figure 3—figure supplement 1A*). Deletion of residues 64-130, which correspond to the majority of the signalling domain, yet retaining the sequence upstream of the TonB box observed in our crystal structure, had no effect on the constitutive and cooperative binding of TonB$_{Ct}$ to FoxA (*Figure 3—figure supplement 1B*). However, the deletion of residues 64-143, which also include the upstream binding motif of the TonB box, abrogated the constitutive binding of TonB$_{Ct}$ to FoxA, whilst retaining the ability to cooperatively form the ternary complex between FoxA, ferrioxamine B and TonB$_{Ct}$ (*Figure 3—figure supplement 1C*). In combination with our structure of the ternary complex, we propose a two-step binding mechanism: The constitutive mode of TonB$_{Ct}$ binding is mediated by the stretch of amino acids located upstream of the TonB box, and the interaction with ferrioxamine B would result in the allosteric release of the TonB box from within the barrel interior allowing the formation of a very tight complex between ferrioxamine B-bound FoxA and TonB, which is necessary for downstream translocation events leading to siderophore uptake.

## Loop movements establish additional contacts with the bound siderophore

Siderophore capture by a specific TBDT is an integral part of establishing the necessary contacts with the TonB/ExbB/ExbD complex, which provides the energy for substrate translocation. As we have demonstrated, the full engagement of TonB by FoxA is dependent on the presence of ferrioxamine B. To understand how FoxA interacts with its substrate ferrioxamine B, we co-crystallised and determined the structure of FoxA with ferrioxamine B in absence of TonB$_{Ct}$ and compared this structure with apo FoxA as well as the ternary complex. Our ferrioxamine B–bound structures of FoxA revealed a common siderophore binding mode similar to other TBDTs. The electron density for the bound ferrioxamine B in both structures allowed us to unambiguously model the conformation of the ligand as judged by the Polder omit maps. (*Figure 4—figure supplement 1A,B*). Ferrioxamine B occupies a highly hydrophobic cavity and is stabilized predominantly through hydrophobic and van der Waals interactions via the aromatic residues found in the surrounding loops and the plug domain inside the cavity (*Figure 4A,B*). Several hydrogen bonds are observed between Tyr$_{805}$ and FoaB$_{O23}$, Tyr$_{218}$ and FoaB$_{N12}$, and His$_{374}$ and FoaB$_{O22}$. The His$_{374}$ residue, located on the 3$^{rd}$ extracellular loop, faces the octahedrally-coordinated Fe(III) and allows the imidazole side chain to form a hydrogen bond with the hydroxamate groups of the siderophore. Additionally, a number of hydrogen bonds are observed between ferrioxamine B and the surrounding water molecules. One of the water molecules involved in the coordination is stabilized through hydrogen bonds by Gln$_{441}$ (*Figure 4B*).

The inward movement of loop 8, which is seen in the ternary complex with TonB$_{Ct}$, places Lys$_{657}$ into close proximity of the bound ferrioxamine B with its ε-amino group protruding towards the hydroxamate groups coordinating Fe$^{3+}$ on the opposite side of His$_{374}$ (*Figure 4C*). These interactions presumably enforce the directionality of siderophore passage in the instance where the high-affinity site on the plug domain is modified during the partial unfolding and provides additional stabilising contacts to the siderophore prior to steps leading to its transport through the lumen of the barrel.

We measured the interaction between ferrioxamine B and purified FoxA using tryptophan fluorescence quenching experiments and ITC. Titration of ferrioxamine B to FoxA purified in nonyl glucopyranoside lead to concentration-dependent quenching of Trp fluorescence and allowed us to calculate the dissociation constant ($K_d$) of 100 ± 10 nM, with a 1:1 stoichiometry (*Figure 4—figure supplement 1C*). Our ITC experiments titrating ferrioxamine B into FoxA yielded a $K_d$ value of 210 ± 44 nM, which agrees well with our fluorescence experiments (*Figure 4—figure supplement 1D*). Such strong association is also observed in other widely studied TBDT-siderophore complexes (*Mislin et al., 2006*) and reflects the highly specific nature of these transporters at capturing extremely scarce siderophore-iron chelates from the extracellular milieu. Our crystal structures of apo FoxA and the ferrioxamine B-bound states revealed only minor conformational perturbations of the extracellular loops and the periplasmic side of the plug domain in FoxA upon binding with the siderophore. However, when ligand-loaded, FoxA association with TonB$_{Ct}$ leads to the closure of the extracellular loops, shielding the bound siderophore inside the barrel. It is worth mentioning that the ability of FoxA to bind ferrioxamine B is unaffected by the bound TonB$_{Ct}$ (*Figure 4—figure supplement 2*), meaning that the movement of the loops is not initiated by the binding of TonB to the apo state of the transporter.

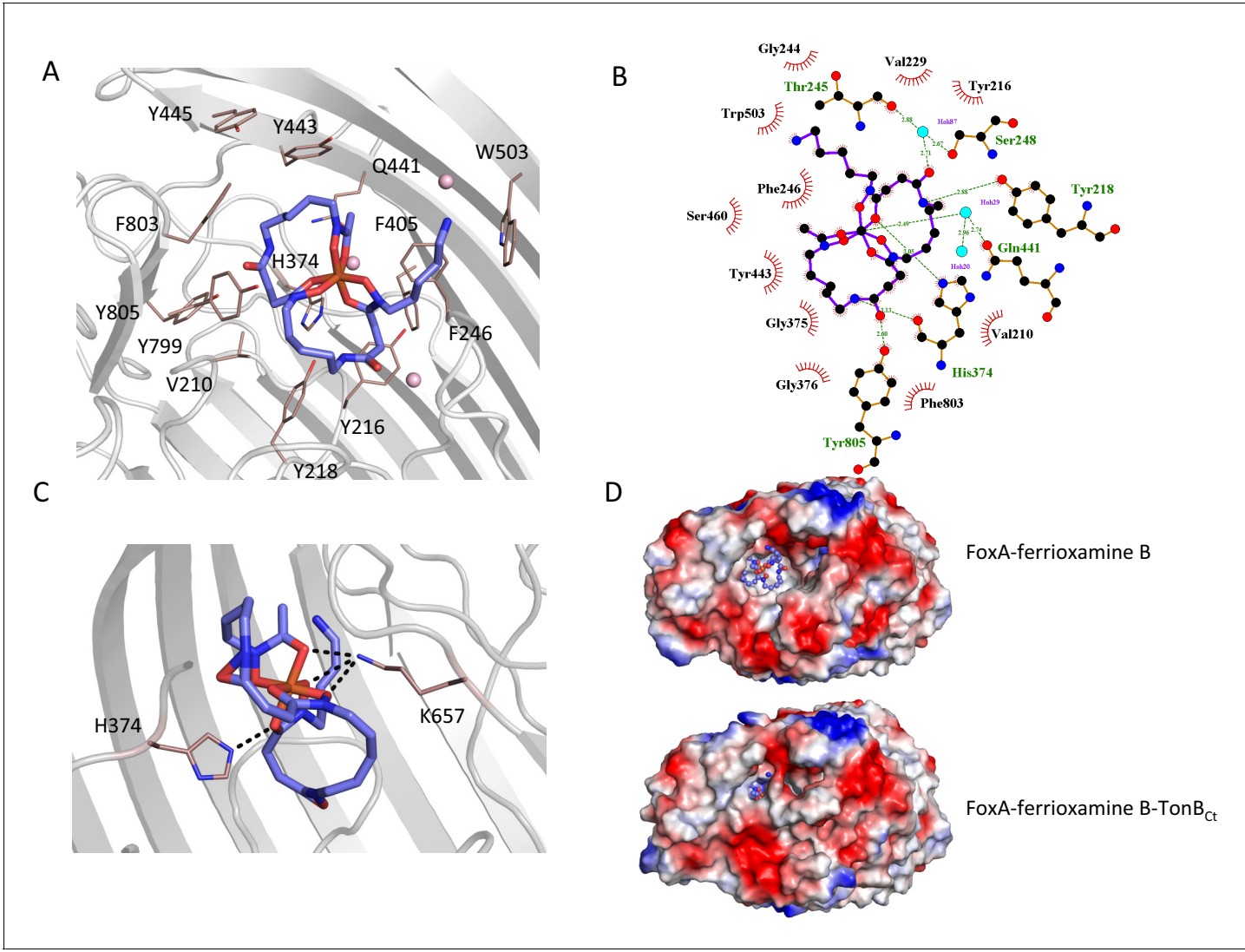

**Figure 4.** Ferrioxamine B interactions with FoxA reveal the basis for high-affinity siderophore capture. (**A**) and **B**) Interaction of ferrioxamine B by hydrophobic and aromatic residues lining the binding pocket of FoxA. Several hydrogen bonds are also observed between ferrioxamine B and residues $His_{374}$ and a $Gln_{441}$-$H_2O$ network, respectively. (**C**) The closure of loop eight results in additional hydrogen bonds between ferrioxamine B and the ε-amino group of $Lys_{657}$ facing the siderophore, which becomes locked from both sides by hydrogen bonds. (**D**) Large hydrophobic cavity facing the extracellular milieu occupied by ferrioxamine B. In the apo/ferrioxamine B structures the cavity and ferrioxamine B are solvent exposed (top), whereas in the ternary complex (bottom) the loop closure sequesters ferrioxamine B inside the barrel.

DOI: https://doi.org/10.7554/eLife.48528.010

The following figure supplements are available for figure 4:

**Figure supplement 1.** Characterisation of ferrioxamine B binding to FoxA.

DOI: https://doi.org/10.7554/eLife.48528.011

**Figure supplement 2.** ITC measurement titrating 500 µM ferrioxamine B into 15–20 µM of pre-assembled FoxA-TonB$_{Ct}$ in nanodiscs shows that constitutive binding of TonB$_{Ct}$ does not lead to the closure of extracellular loops in apo FoxA.

DOI: https://doi.org/10.7554/eLife.48528.012

## Conclusions

Our structures enable us to postulate a mechanistic model for the TonB-mediated uptake of ferrioxamine B by the TBDT FoxA. FoxA can sequester free ferrioxamine B from solution and remain siderophore-bound until it is able to engage with TonB for transport of the siderophore inside the periplasm (**Figure 5**, left half). Likewise, FoxA can form a constitutive complex with TonB, even in the absence of ferrioxamine B. Ferrioxamine B binding to this constitutive state of FoxA-TonB would

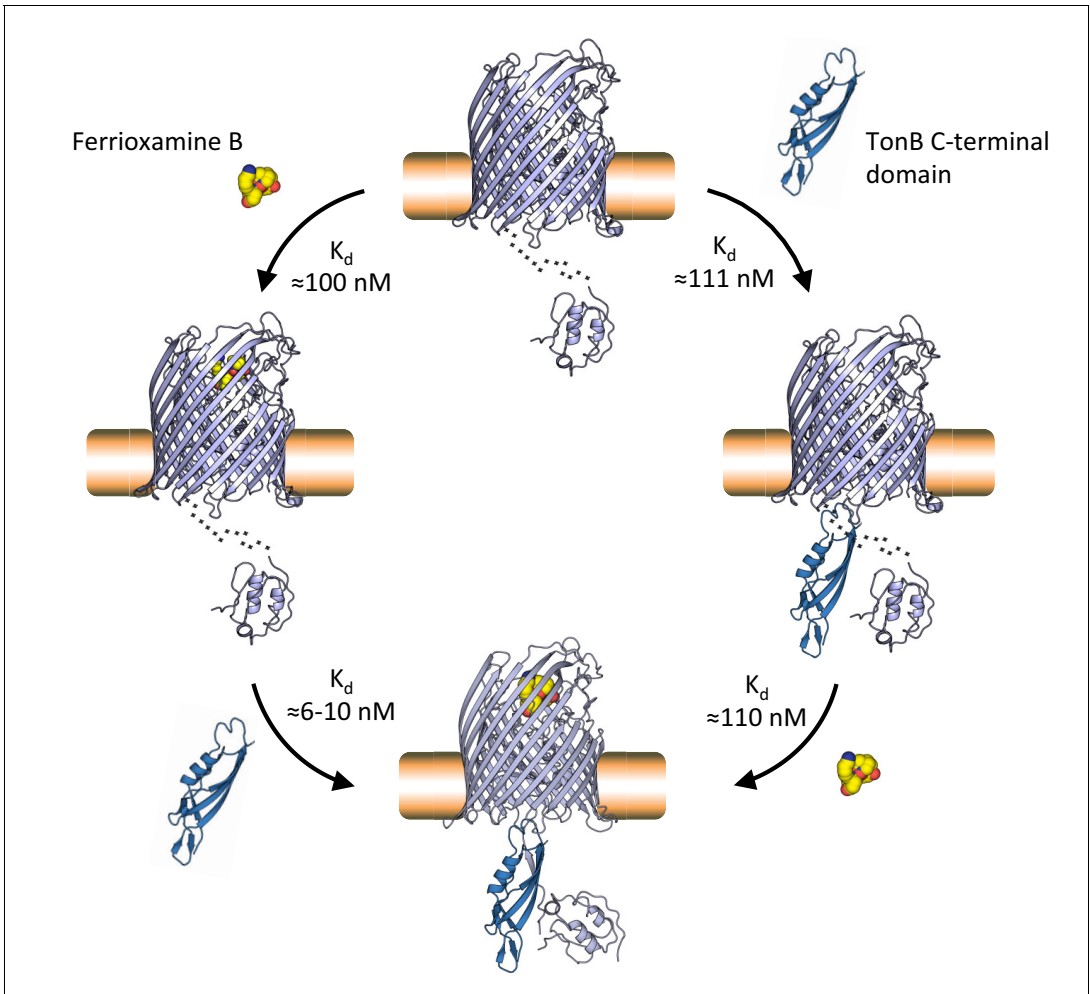

**Figure 5.** Proposed mechanism of TonB-mediated ferrioxamine B uptake via the FoxA transporter. Our studies suggest that FoxA can exist in several states depending on the abundance of ferrioxamine B in the environment and the occupancy of TonB by other TBDRs. FoxA is able to interact with ferrioxamine B or engage with TonB with near-equal affinity ($K_d \approx 100$ nM) in a constitutive fashion. The presence of ferrioxamine B, however, is necessary for the expulsion of the TonB box into the periplasm and the formation of the full, translocation-competent ternary complex through β-augmentation ($K_d \approx 6$–10 nM). This very high-affinity interaction provides the necessary contacts in the complex for subsequent steps of the plug domain re-modelling or expulsion, necessary for siderophore translocation through the lumen of the barrel.
DOI: https://doi.org/10.7554/eLife.48528.013

then lead to the ternary high-affinity complex with TonB via β-augmentation and initiate a cascade of structural re-arrangements within the plug domain to allow for the passage of the substrate through the barrel lumen (*Figure 5*, right panel).

Previous studies into ferrioxamine B uptake suggested that the expression levels of FoxA, compared with other TBDTs like FpvA and FptA, are non-detectable (*Llamas et al., 2006*). Yet, in the presence of ferrioxamine B and under iron-limiting conditions the levels of FoxA in the OM reach those of other TBDTs, which can reach 100-fold the TonB concentration (*Klebba, 2003*). This fact indicates that the "left half" of the proposed mechanism of TonB-mediated ferrioxamine B uptake via the FoxA transporter as shown in *Figure 5* might be the dominating path in vivo under iron-limiting conditions and in the presence of ferrioxamine B. This path is also favoured by the significantly higher affinity of TonB$_{Ct}$ towards "activated" FoxA/FoaB.

The physiological function for the formation of a constitutive FoxA-TonB complex in the absence of ferrioxamine B is currently unclear, but may relate to the basal detection of environmental ferrioxamine B. This would subsequently amplify the *fox* operon leading to efficient uptake of ferrioxamine B from the environment, with little cost of native siderophore production.

Our structure also offers potential insight into signal transduction necessary for the regulation of transcription of the FoxA operon. The signalling domain, visible in the ternary complex, is exposed N-terminally to the periplasmic space, allowing it to engage with the sigma factor regulator protein FoxR situated in the IM. The observed orientation of the signalling domain differs starkly from previous structural studies involving FpvA (*Brillet et al., 2007*; *Wirth et al., 2007*).

# Materials and methods

## Key resources table

| Reagent type (species) or resource | Designation | Source or reference | Identifiers | Additional information |
|---|---|---|---|---|
| Gene (*Pseudomonas aeruginosa*) | FoxA (*Pseudomonas aeruginosa* strain PAO1) | | PA2466 | |
| Strain, strain background (*E. coli*) | Lemo 21 | New England Biolabs | C2528J | |
| Strain, strain background (*E. coli*) | BL21 Gold (DE3) | Agilent | 230132 | |
| Chemical compound, drug | desferrioxamine B | Sigma-Aldrich | D9533 | |
| Chemical compound, drug | octyl glucopyranoside (OG) | Anatrace | O311 | |
| Chemical compound, drug | nonyl glucopyranoside (NG) | Anatrace | N324 | |
| Chemical compound, drug | C8E4 | Anatrace | T350 | |
| Software, algorithm | XDS | (*Kabsch, 2010*) | http://xds.mpimf-heidelberg.mpg.de/ | |
| Software, algorithm | AIMLESS | (*Evans, 2011*) | http://www.ccp4.ac.uk/html/aimless.html | |
| Software, algorithm | Phaser | (*McCoy et al., 2007*) | http://www.ccp4.ac.uk/html/phaser.html | |
| Software, algorithm | PHENIX 1.14 | (*Adams et al., 2010*) | https://www.phenix-online.org/ | |
| Software, algorithm | Coot 0.8.9.1 | (*Emsley et al., 2010*) | https://www2.mrc-lmb.cam.ac.uk/personal/pemsley/coot/ | |
| Software, algorithm | REFMAC5 | (*Murshudov et al., 2011*) | http://www.ccp4.ac.uk/html/refmac5.html | |
| Software, algorithm | Buster-TNT | (*Blanc et al., 2004*) | https://www.globalphasing.com/buster/ | |

## Materials

The detergents used for purification were from Anatrace (Maumee, OH, USA). Desferrioxamine B was purchased from Sigma-Aldrich. All other chemicals were of analytical grade and obtained from Roth (Karlsruhe, Germany) or Sigma Aldrich / Merck (Darmstadt, Germany).

## Protein expression and purification

Full-length FoxA gene from *Pseudomonas aeruginosa* strain PAO1 was cloned into a modified pET28a vector bearing a C-terminal TEV cleavage site prior a His$_6$-tag. Protein overexpression was carried out in *Escherichia coli* Lemo21 cells (*Schlegel et al., 2012*) in 2xTY media supplemented with

NPS (50 mM $Na_2HPO_4$, 50 mM $KH_2PO_4$, 25 mM $(NH_4)2SO_4$) and 5052 mix (0.05% glucose, 0.2% lactose, 0.5% glycerol) and 0.5 mM L-rhamnose. Cells were grown to an $OD_{600}$ of 1 at 37 °C, the temperature was reduced to 20 °C and 0.1 mM IPTG was added for further 16 hours. Cells were lysed in 30 mM Tris pH7.5, 200 mM NaCl, 10% glycerol using the high-pressure homogenizer (EmulsiFlex-C3, Avestin) and cell debris was removed by centrifugation at 22,000 g for 30 min. 1% Triton X-100 was added to the clarified cell lysate and incubated for 1 hr at 4 °C. The outer membrane fraction was isolated by a second centrifugation step at 100,000 g and the resuspended pellet was solubilised overnight in 1% octyl glucopyranoside (OG). Insoluble material was removed by another centrifugation step at 100,000 g for 20 min. Solubilised OM fraction was applied to the Ni-NTA resin followed by subsequent washes with buffer containing 25 mM imidazole and 0.4% nonyl glucopyranoside (NG) or 0.4% C8E4. Protein was eluted with 250 mM imidazole in buffer with 0.4% NG or C8E4 and tobacco etch virus protease (1:10 w/w) was added to the eluted fractions overnight. After reverse Ni-NTA purification, the protein was concentrated and passed over a Superdex S200 10/300 size exclusion column.

$TonB_{Ct}$ (TonB1 from from *Pseudomonas aeruginosa* strain PAO1, residues 251-340) was overexpressed in *Escherichia coli* BL21 Gold cells, grown at 37 °C in LB medium. Cells were induced with 0.2 mM IPTG at $OD_{600}$ of 0.6-0.8 and the temperature was reduced to 20 °C. After 12 hours cells were spun down and lysed in 30 mM Tris pH 7.5, 500 mM NaCl, 10% glycerol using the EmulsiFlex-C3 (Avestin) homogeniser and cell lysate was centrifuged at 40,000 g to remove the cell debris. Cleared cell lysate was supplemented with 20 mM imidazole and loaded onto the 5-ml HisTrap Ni-NTA column. After several washes the protein was eluted with resuspension buffer supplemented with 300 mM imidazole. TEV was added to the pooled fractions containing $TonB_{Ct}$ and reverse-purification was performed the next day to remove the TEV and cleaved $His_6$ tag. $TonB_{Ct}$ was concentrated and stored at -80 °C until further use.

MSP1D1 was expressed and purified as previously described (*Josts et al., 2018*; *Ritchie et al., 2009*) (*Nitsche et al., 2018*). Briefly, MSP1D1 in pET28a vector was transformed in *E. coli* strain BL21 (DE3) and grown in *terrific broth* (TB) media at 37 °C. At an $OD_{600}$ of 1.5 the protein expression was induced by adding 1 mM isopropyl ß-D-1-thiogalactopyranoside (IPTG) and cells were grown for 4 h at 37 °C. Cells were harvested by centrifugation at 3000 g, resuspended in lysis buffer (50 mM Tris pH 8.0, 500 mM NaCl) with 1% Triton X–100 and broken using sonication. The cleared lysate was loaded onto a HisTrap column and washed with ten column volumes each of lysis buffer containing 1% Triton X-100 and 50 mM cholate, respectively. MSP1D1 was eluted with buffer containing 500 mM imidazole, and fractions containing pure protein were pooled and incubated with TEV protease overnight. Subsequently, the protease and cleaved His-tag were separated by applying a second IMAC chromatography step and MSP1D1 without His-tag was concentrated up to 400 μM and stored at -80 °C until further use.

## Analytical size-exclusion chromatography (SEC)

Truncation mutants of FoxA (Δ64-130 and Δ64-143 residues) were generated using the standard QuikChange PCR mutagenesis protocols. Analytical SEC analyses of complex formation between FoxA, truncated FoxA and $TonB_{Ct}$ were performed using a Superdex S200 10/300 column. The buffer in these experiments consisted of 25 mM HEPES pH7.4, 150 mM NaCl, 0.4% NG. In all cases, 15-25 μM FoxA or its truncated forms were mixed with 5-fold excess $TonB_{Ct}$ and 10-fold excess ferrioxamine B (where appropriate) and injected onto the analytical SEC column.

## Crystallisation

Crystals of apo FoxA purified in C8E4 were grown by the sitting drop vapour diffusion technique. 1 μl of purified protein (5-10 mg/ml) was mixed with 1 μl of 1.8-2 M ammonium sulfate solution and 0.1 M HEPES pH 7. Crystals appeared overnight and grew out of phase separation liquid clusters with crystal sizes, limited by the liquid phase, reaching approximately 30-70 μm. For crystallisation of FoxA with ferrioxamine B in NG detergent, the ligand was added in excess to the protein solution and incubated for at least 30 min on ice prior to setting up crystallisation trays. Crystals were cryoprotected by a step-wise addition of glycerol to a final concentration of 18-20% (v/v). For the crystallisation of the FoxA-FoaB-$TonB_{Ct}$ complex, FoxA-FoaB was incubated with 2-fold excess $TonB_{Ct}$, and passed through the size-exclusion column. Fractions corresponding to the complex were pooled and

set up in crystallisation screens. The complex crystallised in the same condition as the apo FoxA and FoxA-FoaB, containing 1.8-2.0 M ammonium sulfate, 0.1 M HEPES pH 7. These crystals were cryo-protected by soaking the crystals in the crystallisation condition supplemented with 20% ethylene glycol for 2-3 min.

## Structure determination

All X-ray diffraction data were collected at 100 K. Data were collected at the PETRA III/EMBL P14, ESRF ID30B and BESSY 14.1 beamlines. All datasets were processed with XDS (*Kabsch, 2010*) and merged with AIMLESS (*Evans, 2006*; *Evans, 2011*). All final data were merged from two individual datasets. Unit cell parameters and space groups are given in Table 1 (*Supplementary file 1*). We used the FhuA model from *Escherichia coli* (PDB:1BY3) as a molecular replacement candidate (40-45% sequence identity to FoxA). After the successful placement of the model using Phaser (*McCoy et al., 2007*), the FoxA model was completed using a combination of phenix.autobuild (*Terwilliger et al., 2008*) and manual building in Coot (*Emsley and Cowtan, 2004*). Apo FoxA was refined using phenix.refine and REFMAC5 (*Afonine et al., 2012*) (*Murshudov et al., 2011*). For the FoxA-FoaB complex, refinement was performed initially using phenix.refine, followed by TLS and jelly-body refinement in REFMAC5 (*Murshudov et al., 2011*). For modelling the FoxA-FoaB-TonB$_{Ct}$ complex, apo FoxA was used as a search model in Phaser. Once the MR solution was identified, clear density corresponding to the TonB$_{Ct}$ and the N-terminal signalling domain became evident and they were manually built into the electron density using Coot. Refinement was carried out initially using phenix.refine at the early stages of model building. Once all the backbone poly-Ala stretches were built, Buster-TNT (*Blanc et al., 2004*) was used for all the subsequent refinement procedures. The final models correspond to residues 44-820 of FoxA with residues 119-124 and 138-155 being disordered. The TonB$_{Ct}$ model comprises residues 251-340. All data collection and refinement statistics are summarised in Table I (*Supplementary file 1*).

## Isothermal titration calorimetry (ITC)

FoxA was incorporated into MSP1D1 nanodiscs. Briefly, FoxA was mixed with purified MSP1D1 and POPC lipids in 1:2:70 ratio and biobeads were added to initiate the nanodisc assembly. After approximately 4-5 hours, the mixture was concentrated and purified by gel size exclusion chromatography using a Superdex S200 10/300 column. All proteins were extensively dialysed against 20 mM HEPES, 150 mM NaCl, pH 7.5 overnight. 15 µM FoxA or FoxA-FoaB incorporated into MSP1D1 nanodiscs were loaded into the ITC cell, 150 µM TonB$_{Ct}$ was placed in the syringe. For the "pre-loaded" FoxA-FoaB complex, FoxA in nanodiscs was dialyzed against 1 mM FoaB. All binding reactions were measured at 26 ˚C. TonB$_{Ct}$ was also titrated against empty nanodiscs as a control. Heat of dilution was obtained by titrating TonB$_{Ct}$ into the dialysis buffer and subtracted from all subsequent measurements. Heats of binding for all the reactions were integrated using Microcal Origin software and all the data were fitted to a single-site binding model. All titrations were performed as triplicates and errors are reported as standard deviations (SD).

## Tryptophan fluorescence quenching experiments

All fluorescence measurements were performed using a Cary Eclipse fluorescence spectrometer. FoxA purified in nonyl glucopyranoside (NG) was diluted to 100 nM in a 3 ml quartz cuvette. Trp fluorescence was excited at 280 nm and emission spectra were recorded from 310-420 nm. Ferrioxamine B, diluted in the same buffer as FoxA, was titrated into the cuvette until saturation in fluorescence quenching was reached. Control experiments with buffer were performed to account for dilution effects on Trp fluorescence. Buffer conditions were 20 mM Tris pH 7.5, 200 mM NaCl, 0.4% NG. Curves were plotted and analysed in GraphPad Prism 7. The binding curve was fitted to a single-site binding model. Measurements were performed as triplicates.

## Acknowledgements

We are grateful to the staff at beamlines P13 and P14 (EMBL, Hamburg), BL14.1 (BESSY, Berlin) and ID30B (ESRF, Grenoble) and thank members of the Tidow lab for helpful discussions. We acknowledge access to the Sample Preparation and Characterisation (SPC) Facility of EMBL Hamburg. This research was funded by the excellence cluster 'The Hamburg Centre for Ultrafast Imaging -

Structure, Dynamics and Control of Matter at the Atomic Scale' of the Deutsche Forschungsgemein-schaft (DFG EXC 1074).

## Additional information

### Funding

| Funder | Grant reference number | Author |
|---|---|---|
| Deutsche Forschungsge-meinschaft | DFG EXC 1074 | Henning Tidow |

The funders had no role in study design, data collection and interpretation, or the decision to submit the work for publication.

### Author contributions

Inokentijs Josts, Conceptualization, Investigation, Writing—original draft, Writing—review and editing; Katharina Veith, Investigation; Henning Tidow, Supervision, Funding acquisition, Writing—original draft, Writing—review and editing

### Author ORCIDs

Inokentijs Josts https://orcid.org/0000-0002-8235-2397
Henning Tidow https://orcid.org/0000-0002-4702-9332

### Decision letter and Author response

Decision letter https://doi.org/10.7554/eLife.48528.023
Author response https://doi.org/10.7554/eLife.48528.024

## Additional files

### Supplementary files

• Supplementary file 1. Supplementary tables.
DOI: https://doi.org/10.7554/eLife.48528.014

• Transparent reporting form
DOI: https://doi.org/10.7554/eLife.48528.015

### Data availability

Structural coordinates and structure factors have been deposited in the RCSB Protein Data Bank under accession numbers 6I96, 6I97, and 6I98 (see Table 1, Supplementary file 1).

The following datasets were generated:

| Author(s) | Year | Dataset title | Dataset URL | Database and Identifier |
|---|---|---|---|---|
| Josts I, Tidow H | 2019 | Crystal structure of outer membrane protein | http://www.rcsb.org/structure/6I96 | RCSB Protein Data Bank, 6I96 |
| Josts I, Tidow H | 2019 | Crystal structure of outer membrane protein | http://www.rcsb.org/structure/6I97 | RCSB Protein Data Bank, 6I97 |
| Josts I, Tidow H | 2019 | Crystal structure of outer membrane protein | http://www.rcsb.org/structure/6I98 | RCSB Protein Data Bank, 6I98 |

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
