## [Decision Letter]

Thank you for submitting your article "Crystal structures of the outer membrane transporter FoxA provide novel insights into TonB-mediated siderophore uptake" for consideration by *eLife*. Your article has been reviewed by two peer reviewers, including Volker Dötsch as the Reviewing Editor and Reviewer #1, and the evaluation has been overseen by Olga Boudker as the Senior Editor. The following individual involved in review of your submission has agreed to reveal their identity: Michael Wiener (Reviewer #2).

The reviewers have discussed the reviews with one another and the Reviewing Editor has drafted this decision to help you prepare a revised submission.

Summary:

This is a solid structural biological and biophysical manuscript on the *Pseudomonas* outer membrane (OM) TonB-dependent transporter FoxA, which will be of great interest to those who work on this still-enigmatic class of active transport membrane proteins. These transporters utilize the pmf of the inner/plasma membrane to transport substrates across the outer membrane by using the TonB protein to couple to the ExbB/ExbD motor complex in the plasma membrane. The most striking structural result, compared to what already exists in the literature, is the crystal structure of the ternary complex of FoxA/ferrioxamineB/TonB C-terminal domain. In contrast to the two previous transporter/substrate/TonB complex structures, FoxA contains an N-terminal extension which plays a role in transcriptional regulation of siderophore synthesis. In these previous structures (FhuA and BtuB), the β-sheet of TonB is approximately perpendicular to the β-strands of the transporter barrel, and interacting with both the transporter's "Ton box" and the β-barrel. In contrast, in the FoxA ternary complex, the TonB β-sheet is approximately parallel to the strands of the β-barrel, "pulled away" from the barrel, and interacting with the transporter's Ton-box and residues upstream of the Ton-box, in the linker region between Ton-box and N-terminal extension. This "linker interaction" has not been previously observed structurally. Additional biophysical experiments, primarily ITC but also (intrinsic) fluorescence quenching, as well as SEC of various modified/truncated constructs, indicate that TonB's affinity for FoxA increases almost 20-fold with substrate (ferrioxamineB) bound to FoxA.

Essential revisions:

1) How do these results bear upon the functional mechanism in the bacteria? Is the statement that "The FoxA transporter can form a constitutive complex with TonB, even in the absence of ferrioxoamine B" necessarily true in vivo? This depends on relative concentrations of TonB, FoxA, and "competition" for TonB by the multiple other TonB-dependent transporters present in the OM. These questions should at least be discussed in the manuscript. If data concerning concentrations are available they should also be included.

2) Since TonB binds to apo FoxA with ~100nM K_d_, were any crystallization experiments attempted with this binary complex?

3) The error propagation in the ITC experiments is not clear. For example, how can a 75% fractional error in K_d_ lead to only a 2% fractional error in ΔH (and no stated error for TΔS)? What was the number of replicates?

4) Likewise, errors are missing for the Trp quenching experiments. This should be provided.

5) For ITC determination of TonB binding to FoxA/ferrioxamineB, FoxA appears to have been "pre-loaded" with ferrioxamine B. Please provide more information on how this was done, and how estimated occupancy could affect the observed results. For example, given an ITC determination of ferrioxamine K_d_ for FoxA of 180 +/- 140 nM (78% fractional error), a 1:1 complex of FoxA:ferrioxamineB at 15uM will be ~90% occupancy, so the determined thermodynamic parameters are a combination of +/- substrate-bound FoxA.

6) The term "novel insights" should be eliminated from the title (and elsewhere in the text). Instead the emphasis should be more on this being the first ternary structure (transporter/substrate/TonB[C-terminal periplasmic domain]) for a transporter with an N-terminal signaling domain – and a novel orientation of TonB.

---

## [Author Response]

Essential revisions:1) How do these results bear upon the functional mechanism in the bacteria? Is the statement that "The FoxA transporter can form a constitutive complex with TonB, even in the absence of ferrioxoamine B" necessarily true in vivo? This depends on relative concentrations of TonB, FoxA, and "competition" for TonB by the multiple other TonB-dependent transporters present in the OM. These questions should at least be discussed in the manuscript. If data concerning concentrations are available they should also be included.

These are very valid questions that are difficult to answer given the large number of variables in this system. Regarding the relative concentrations of TonB and FoxA, it is known that FoxA expression could only be observed under iron-restricted conditions in the presence of ferrioxamine B (Llamas et al., 2006). As the FoxA gene contains a Fur box in its promoter region, the expression of FoxA is most likely repressed via regulation by the Fur repressor protein under iron-rich conditions (Llamas et al., 2006). Thus, under resting conditions the TonB concentration should always dominate over the FoxA levels.

On the other hand, if FoxA gets upregulated via a signaling cascade involving binding of ferrioxamine B to FoxA, signal transmission to FoxR (anti-sigma factor) and increased activity of FoxI (sigma factor), its total concentration (like that of many other TBDT) can reach 100-fold of that of TonB (Klebba, 2003). This results in two populations of TBDT in the outer membrane: active transporters associated with TonB and inactive transporters unassociated with it (Newton, Trinh, and Klebba, 2010, J Biol Chem). Whether the “activated” ferrioxamine B-bound FoxA then indeed binds to TonB depends on the relative affinities of TonB to the different TBDT (which seem to be very similar) and their expression levels. In the presence of ferrioxamine B, the production of other siderophores is downregulated (Galet et al., 2015), which would result in fewer “active” TBDTs other than FoxA and thus reduced competition by TonB-binding to other TBDTs.

We have now revised the manuscript and in particular the conclusion section to discuss these aspects in more details. Given the above-mentioned facts, the “left half” of the proposed mechanism of TonB-mediated ferrioxamine B uptake via the FoxA transporter as shown in Figure 5 might be the dominating path in vivo under iron-limiting conditions and in the presence of ferrioxamine B. This path is also favored by the significantly higher affinity of TonB_Ct_ to “activated” FoxA/FoaB (compared to all other affinities along the reaction circle).

2) Since TonB binds to apo FoxA with ~100nM K_d_, were any crystallization experiments attempted with this binary complex?

We have indeed tried very hard to crystallize the binary FoxA/TonB_Ct_-complex as this would be the last missing puzzle piece in the variety of structural states adopted by FoxA during TonB-mediated ferrioxamine B uptake. Unfortunately, despite very pure and stable samples, we never obtained any crystals.

3) The error propagation in the ITC experiments is not clear. For example, how can a 75% fractional error in K_d_ lead to only a 2% fractional error in ΔH (and no stated error for TΔS)? What was the number of replicates?

In the previous version of the manuscript, the reported errors were fitting errors for a single measurement each, and thus quite large. We have now repeated all measurements to obtain triplicates and report the errors as standard deviations (SD) obtained from averaging K_a_-values and converting them to K_d_-values with error propagation. Errors for all other thermodynamic parameters are also given as SD now and included in Table 2 in Supplementary file 1. We have included the information about number of replicates and error propagation in the Materials and methods section now.

4) Likewise, errors are missing for the Trp quenching experiments. This should be provided.

As stated above, we have now performed all binding experiments in triplicates and reported the dissociation constant (K_d_) as average ± SD. For the Trp quenching experiments shown in Figure 4—figure supplement 1C, this results in a K_d_ of 100 ± 10 nM.

5) For ITC determination of TonB binding to FoxA/ferrioxamineB, FoxA appears to have been "pre-loaded" with ferrioxamine B. Please provide more information on how this was done, and how estimated occupancy could affect the observed results. For example, given an ITC determination of ferrioxamine K_d_ for FoxA of 180 +/- 140 nM (78% fractional error), a 1:1 complex of FoxA:ferrioxamineB at 15uM will be ~90% occupancy, so the determined thermodynamic parameters are a combination of +/- substrate-bound FoxA.

The reviewer is correct. For the ITC titration of TonB into FoxA/FoaB, we “pre-loaded” FoxA with ferrioxamine B by dialyzing FoxA in nanodiscs against 1 mM (excess) FoaB. We have added this information to the Materials and methods section now.

Given a K_d_ of 100-200 nM for the binding of FoaB to FoxA (from fluorescence or ITC measurements, respectively) this excess of FoaB during the “pre-formation” of the FoxA/FoaB complex ensures that > 99.9% of FoxA is substrate-bound. This enables a clear distinction of binding parameters between binding of TonB to apo FoxA vs. TonB-binding to the FoxA/FoaB complex.

The errors of all ITC measurements have been updated according to our new triplicate repeat measurements (see Table 2 in Supplementary file 1). Initially, we reported fitting errors of single experiments. Now we report SD errors from triplicate titrations, which result in much lower fractional errors.

6) The term "novel insights" should be eliminated from the title (and elsewhere in the text). Instead the emphasis should be more on this being the first ternary structure (transporter/substrate/TonB[C-terminal periplasmic domain]) for a transporter with an N-terminal signaling domain – and a novel orientation of TonB.

We followed the suggestion of the reviewer/editor and eliminated the term “novel insights”. We have changed the title to “Ternary structure of the outer membrane transporter FoxA with resolved signaling domain provides insights into TonB-mediated siderophore uptake”. This title now puts further emphasis on the fact that the structure presented in this manuscript is the first ternary structure of a TBDT including a signalling domain.

This aspect as well as the different orientation of the TonB_Ct_ (compared to previous studies) has also been emphasized in the Results section:

“Overall, the structure of the ternary FoxA-ferrioxamineB-TonB_Ct_ complex presented in this work reveals both the structure of the N-terminal signalling domain as well as a markedly different orientation of the TonB_Ct_ relative to the TBDT compared to previously determined ternary structures of FhuA and BtuB (Pawelek, Croteau et al., 2006, Shultis, Purdy et al., 2006).”